# Combination of AID2 and BromoTag expands the utility of degron-based protein knockdowns

Yuki Hatoyama [1,2,6], Moutushi Islam [1,2,6], Adam G Bond [3], Ken-ichiro Hayashi[4], Alessio Ciulli [3] & Masato T Kanemaki [1,2,5] ✉

## Abstract

**Acute protein knockdown is a powerful approach to dissecting protein function in dynamic cellular processes. We previously reported an improved auxin-inducible degron system, AID2, but recently noted that its ability to induce degradation of some essential replication factors, such as ORC1 and CDC6, was not enough to induce lethality. Here, we present combinational degron technologies to control two proteins or enhance target depletion. For this purpose, we initially compare PROTAC-based degrons, dTAG and BromoTag, with AID2 to reveal their key features and then demonstrate control of cohesin and condensin with AID2 and BromoTag, respectively. We develop a double-degron system with AID2 and BromoTag to enhance target depletion and accelerate depletion kinetics and demonstrate that both ORC1 and CDC6 are pivotal for MCM loading. Finally, we show that co-depletion of ORC1 and CDC6 by the double-degron system completely suppresses DNA replication, and the cells enter mitosis with single-chromatid chromosomes, indicating that DNA replication is uncoupled from cell cycle control. Our combinational degron technologies will expand the application scope for functional analyses.**

**Keywords** AID2; BromoTag; dTAG; Degron; PROTAC and Protein Degradation
**Subject Categories** Methods & Resources; Post-translational Modifications & Proteolysis

## Introduction

Studies of protein function in living cells are greatly helped by conditional loss-of-function experiments. For this purpose, siRNA-based mRNA depletion and Cre-based conditional knockout have been employed for decades (Elbashir et al, 2001; Gu et al, 1993). However, target-protein depletion by these technologies is relatively slow because it depends on the protein's half-life. For studying dynamic cellular processes such as the cell cycle, gene regulation, and differentiation, rapid depletion of the target protein is crucial to capture the primary defect before the accumulation of secondary defects (Jaeger and Winter, 2021; Kanemaki, 2022). All eukaryotic cells are equipped with the ubiquitin-proteasome system (UPS), which degrades proteins within a few minutes to hours (Kleiger and Mayor, 2014). Therefore, protein knockdown by a conditional degron through the UPS provides an ideal methodology to study the rapid depletion of target proteins in living cells.

We previously developed one of the major conditional degron systems, auxin-inducible degron (AID), by transplanting a plant-specific degradation pathway into non-plant cells (Nishimura et al, 2009). For inducing target degradation, a degron tag derived from *Arabidopsis thaliana* IAA17 (e.g., mini-AID (mAID) and AID*) is fused to a protein of interest in cells expressing F-box protein *Oryza sativa* TIR1 (OsTIR1), which forms a SKP1–CUL1–F-box (SCF) E3 ligase with the endogenous components (Morawska and Ulrich, 2013; Natsume et al, 2016). A natural auxin, indole-3-acetic acid (IAA), added to the culture medium binds to OsTIR1 and, subsequently, IAA-bound SCF–TIR1 recognizes and poly-ubiquitylates the degron for rapid degradation via the UPS. The original AID system showed a leaky degradation by SCF–OsTIR1 even without IAA addition, and the concentrations of IAA required for inducing target degradation were high (100 to 500 μM), which potentially exhibits toxicity in some cells. We and others recently overcame these problems by establishing an AID version 2 (AID2) system, in which we employed an OsTIR1(F74G/A) mutant and an auxin analog, 5-Ph-IAA (Fig. 1A; Appendix Fig. S1A) (Nishimura et al, 2020; Yesbolatova et al, 2020). We have successfully shown that AID2 allowed us to degrade proteins involved in DNA replication for phenotypic analyses (Klein et al, 2021; Lim et al, 2023; Liu et al, 2023; Saito et al, 2022). However, we recently noted that there were instances where target depletion by AID2 was not sufficient for phenotypic studies. In this study, we sought to establish a methodology to enhance target depletion by AID2.

Recently, other conditional degron technologies have been developed based on immunomodulatory drugs (IMiDs) or proteolysis-targeting chimeras (PROTACs) that utilize an endogenous E3 ligase such as CRL4–CRBN and CRL2–VHL

[1]Department of Chromosome Science, National Institute of Genetics, Research Organization of Information and Systems (ROIS), Yata 1111, Mishima, Shizuoka 411-8540, Japan. [2]Graduate Institute for Advanced Studies, SOKENDAI, Yata 1111, Mishima, Shizuoka 411-8540, Japan. [3]Centre for Targeted Protein Degradation, School of Life Science, University of Dundee, 1 James Lindsay Place, Dundee DD1 5JJ Scotland, UK. [4]Department of Biochemistry, Okayama University of Science, Ridai-cho 1-1, Okayama 700-0005, Japan. [5]Department of Biological Science, Graduate School of Science, The University of Tokyo, Bunkyo-ku, Tokyo 113-0033, Japan. [6]These authors contributed equally: Yuki Hatoyama, Moutushi Islam. ✉E-mail: mkanemak@nig.ac.jp

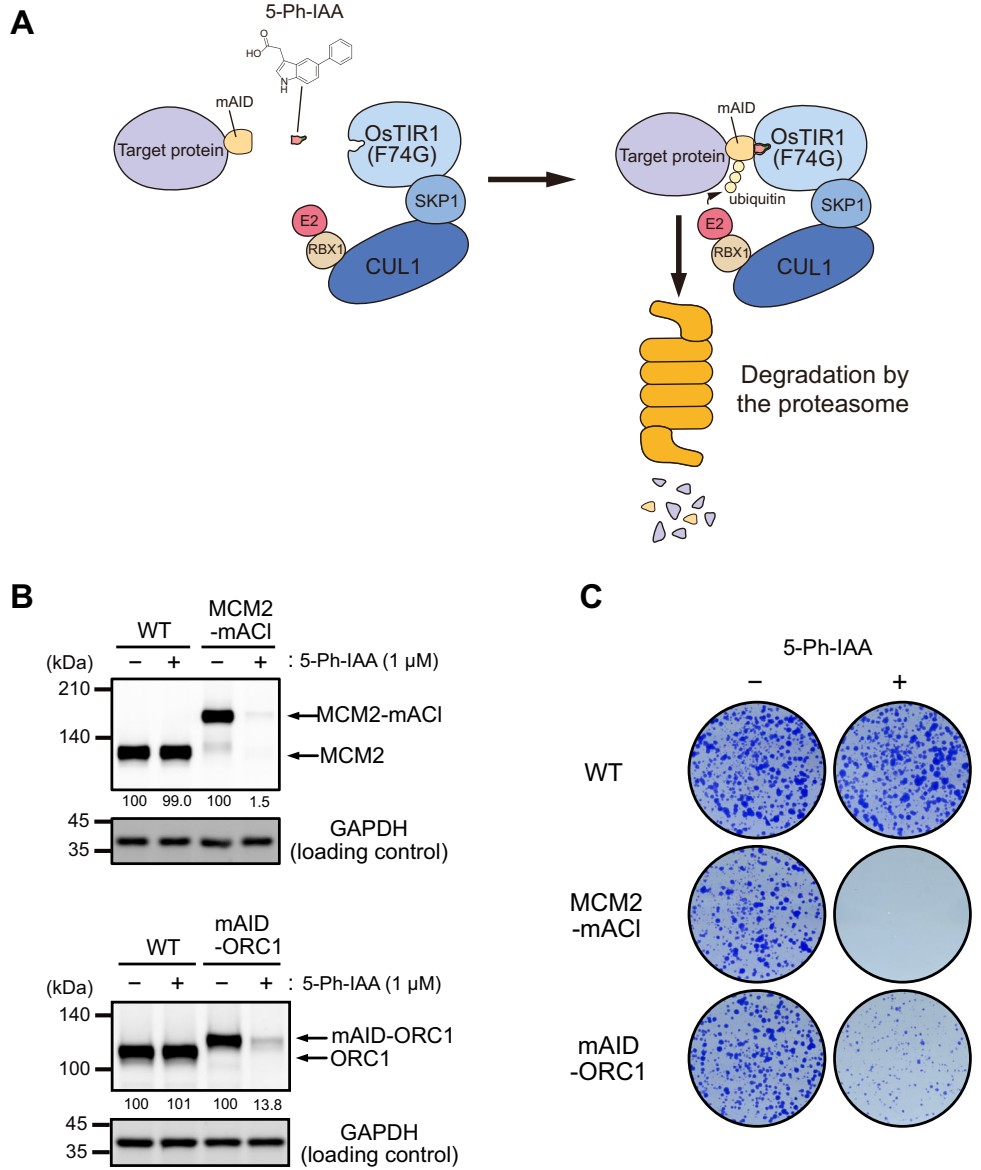

**Figure 1. AID2-mediated depletion of MCM2 and ORC1 in HCT116.**

(**A**) Schematic illustration of the targeted protein degradation by the AID2 system. (**B**) Depletion of MCM2-mAID-mClover (MCM2-mACl) and mAID-ORC1. The parental HCT116 wild-type (WT), MCM2-mACl and mAID-ORC1 cells were treated with 1 μM 5-Ph-IAA for 6 h. Relative MCM2 and ORC1 levels taking the DMSO-treated control as 100% are shown under each blot. Each data point was normalized with the corresponding GAPDH loading control. Proteins were detected by anti-MCM2, -GAPDH, and -tubulin antibodies. (**C**) Colony formation of the parental HCT116 WT, MCM2-mACl, and mAID-ORC1 cells. The indicated cells were cultured in the presence or absence of 1 μM 5-Ph-IAA for 7 days. Colonies were stained with crystal violet. Source data are available online for this figure.

(Bond et al, 2021; Bouguenina et al, 2023; Buckley et al, 2015; Koduri et al, 2019; Nabet et al, 2018; Nowak et al, 2021; Yamanaka et al, 2020). Because the original IMiDs without recent improvements inevitably induce off-target proteolysis of CRBN neosubstrates (Mercer et al, 2024), PROTAC-based technologies have been an attractive choice for many researchers. Among PROTAC-based degrons, dTAG technology employs a PROTAC degrader, dTAG-13 or dTAGv-1, and FKBP12(F36V) as a degron (Appendix Fig. S1A) (Nabet et al, 2020; Nabet et al, 2018). A protein fused with FKBP12(F36V) (hereafter the degron tag is also called dTAG) is recruited to CRL4–CRBN or CRL2–VHL in the presence

of dTAG-13 or dTAGv-1, respectively, for rapid degradation. BromoTag and a similar BRD4-based degron were recently reported (Bond et al, 2021; Nowak et al, 2021). BromoTag utilizes a PROTAC degrader AGB1 and an L387A mutant derived from the second BRD4 bromodomain as a degron (hereafter, the degron tag is also called BromoTag) (Appendix Fig. S1A). A protein fused with BromoTag is recruited to CRL2–VHL in the presence of AGB1 for rapid proteolysis via the UPS. Both dTAG and BromoTag have been used in many studies, indicating that they are also promising conditional degrons (Evrin et al, 2023; Mylonas et al, 2021; Olsen et al, 2022; Weintraub et al, 2017). However, few

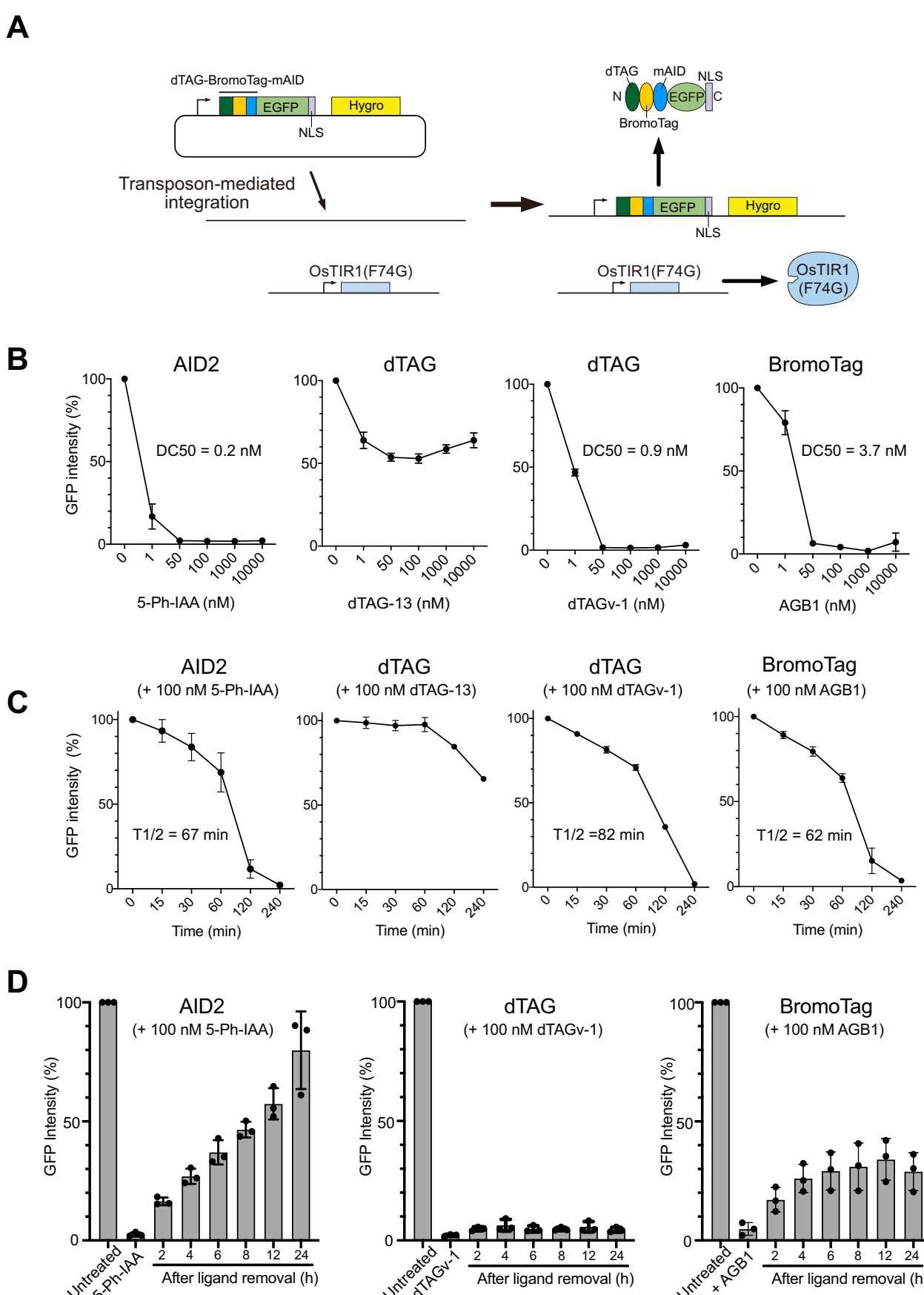

**Figure 2.  Comparing the AID2, dTAG, and BromoTag systems using the reporter HCT116 cells.**

(A) Schematic illustration showing the strategy to generate the reporter HCT116 cell line expressing OsTIR1(F74G) and the GFP reporter. (B) Dose-response of reporter depletion. The reporter cells were treated with the indicated concentrations of each ligand for 4 h. GFP intensity was analysed, taking the mock-treated cells as 100%. Data were presented as mean ± SD of four technical replicates ($n = 4$). The DC50 values were calculated with the nonlinear regression model on GraphPad Prism 8. (C) Time-course depletion of the reporter. The reporter cells were treated with 100 nM of the indicated ligand. Samples were taken at the indicated time points, and the GFP intensity was analysed taking the mock-treated cells as 100%. Data were presented as mean ± SD of four technical replicates ($n = 4$). The T1/2 was calculated with the nonlinear regression model on GraphPad Prism 8. (D) Re-expression of the reporter after depletion by the AID2, dTAG, or BromoTag system. The reporter cells were treated with 100 nM 5-Ph-IAA, dTAGv-1, or AGB1 for 4 h before the medium change. Samples were taken at the indicated time points, and the GFP intensity was analysed taking the mock-treated cells as 100%. Data were presented as mean ± SD. Each dot represents a technical replicate ($n = 3$). Source data are available online for this figure.

studies have compared conditional degrons to discern their similarities and differences, and BromoTag has not yet been included for comparison (Bondeson et al, 2022; Noviello et al, 2023). Furthermore, the target proteins fused with each degron in these studies were overexpressed to varying levels, impacting degradation kinetics and complicating the comparison. Ideally, a comparison of different degron systems should be conducted with a cell line expressing the same level of target protein.

To achieve the goal of enhancing target depletion by AID2, we proposed combining AID2 with another degron system, as others previously showed using AID and SMASh (Crncec and Hochegger, 2022; Lemmens et al, 2018). We initially compared AID2, dTAG, and BromoTag using a single GFP reporter containing three degron tags in tandem to understand their similarities and differences. We found that all degron systems achieve rapid reporter depletion. However, AID2 exhibits superior performance in terms of depletion efficiency, kinetics, and reversible expression recovery in HCT116 and hTERT-RPE1 cells. Subsequently, we showed that two proteins can be independently and simultaneously depleted using AID2 and BromoTag. Finally, we showed that a double-degron system with mAID-BromoTag enhances target depletion, accelerates depletion kinetics, and confers strong phenotypic defects. By using this double-degron system, we succeed in showing that both ORC1 and CDC6 are pivotal for the MCM-loading prerequisite for DNA replication. Furthermore, we achieved complete suppression of DNA replication by co-depleting ORC1 and CDC6, leading cells to enter mitosis without DNA replication.

## Results

### AID2-mediated ORC1 depletion does not result in a strong growth defect

We previously established an improved version of the auxin-inducible degron (AID) system, namely AID2, and reported that degron-fused proteins were sharply induced for rapid degradation after the addition of 5-Ph-IAA (Fig. 1A) (Yesbolatova et al, 2020). Because our group is interested in DNA replication, we applied AID2 to control the expression of essential replication factors in human colorectal cancer HCT116 cells by biallelically tagging the endogenous target genes using CRISPR-Cas9 (Lim et al, 2023; Liu et al, 2023; Saito et al, 2022). When we applied AID2 to the MCM2 subunit of the replicative MCM2–7 helicase, we successfully depleted MCM2 C-terminally fused with mAID-mClover (MCM2-mACl) upon the addition of 5-Ph-IAA (Fig. 1B, upper blots). Consequently, MCM2-mACl cells treated with 5-Ph-IAA stopped

growing and did not form any colonies, suggesting that they did not carry out DNA replication (Fig. 1C).

ORC1 is a subunit of the ORC1–6 complex, which plays a pivotal role in loading MCM2–7 to chromosomal DNA, and loss of ORC1 causes lethality in budding yeast (Klemm and Bell, 2001). However, there are conflicting reports on whether ORC1 is essential for DNA replication or not in human cells (Chou et al, 2021; Shibata et al, 2016). To clarify this issue, we generated ORC1-degron cells with AID2 by fusing mAID to the N-terminus of ORC1. The mAID-ORC1 protein was efficiently depleted upon the addition of 5-Ph-IAA (Fig. 1B, lower blots). Unexpectedly, mAID-ORC1 cells grew slowly and formed small colonies, suggesting that ORC1-depleted cells carried out DNA replication even though the cells were defective (Fig. 1C). We interpreted this result as suggesting that ORC1 was likely to be required for human DNA replication, but the depletion by AID2 was not enough to cause lethality, and the required amount of ORC1 for DNA replication was very low compared with that of MCM2. We found a similar case with another replication factor, CDC6 (described later). These findings motivated us to enhance depletion by the AID2 system.

### Understanding similarities and differences of AID2, dTAG, and BromoTag

To overcome the challenges mentioned above, we sought to combine AID2 and another degron system. We became interested in the recently reported PROTAC-based degrons, dTAG and BromoTag (Appendix Fig. S1A) (Bond et al, 2021; Nabet et al, 2020; Nabet et al, 2018; Nowak et al, 2021). Even though AID2, dTAG and BromoTag have been used in cell biology, there are only a few studies comparing these systems (Bondeson et al, 2022; Noviello et al, 2023). To understand the similarities and differences among AID2, dTAG, and BromoTag systems, we constructed a GFP reporter containing the three degrons in tandem (Fig. 2A). These degrons (dTAG, BromoTag, and mAID, respectively) were connected with a long linker (11–12 amino-acid chain composed of G, A and S) to ensure that they can each be readily accessed by the corresponding E3 ubiquitin ligase enzyme complex. In order to focus on nuclear proteins such as ORC1, we also fused a nuclear localization signal (NLS) to the C-terminus and confirmed its nuclear localization (Appendix Fig. S2A). We transfected a transposon plasmid encoding the reporter gene to parental HCT116 wild-type (WT) cells expressing OsTIR1(F74G) from the safe harbor *AAVS1* locus and selected stable cells as previously reported (Fig. 2A) (Yesbolatova et al, 2020).

We initially tested ligand concentrations required for inducing reporter degradation (Fig. 2B). The reporter cells were treated with a differential concentration of inducing ligand for 4 h, and

**A**

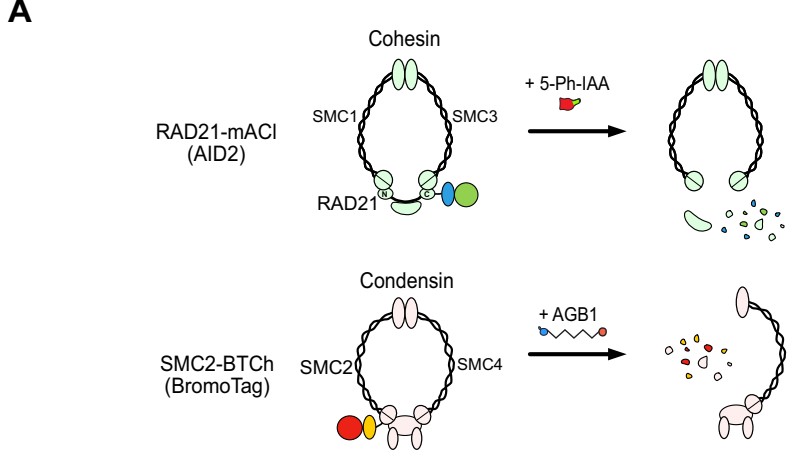

**B**

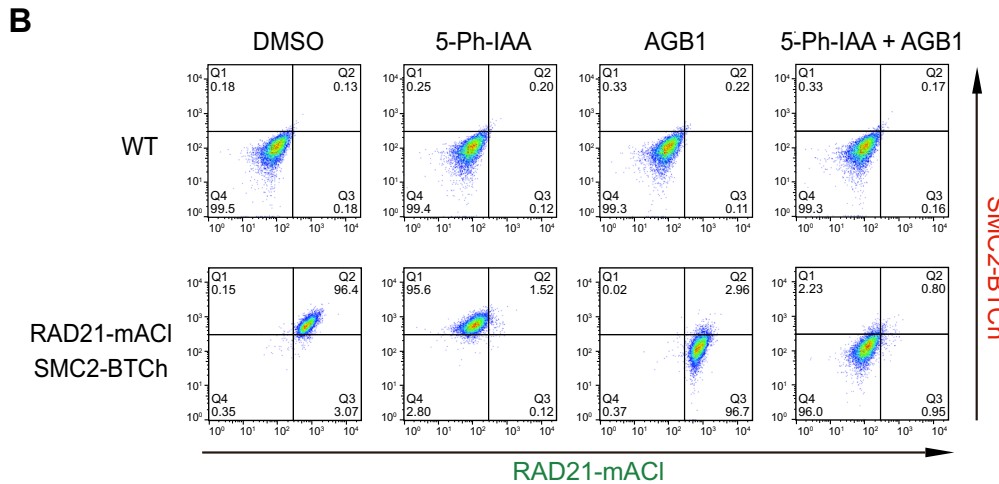

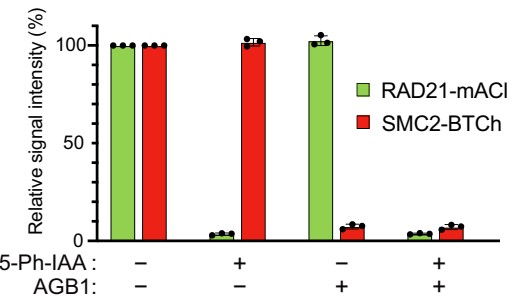

**C**

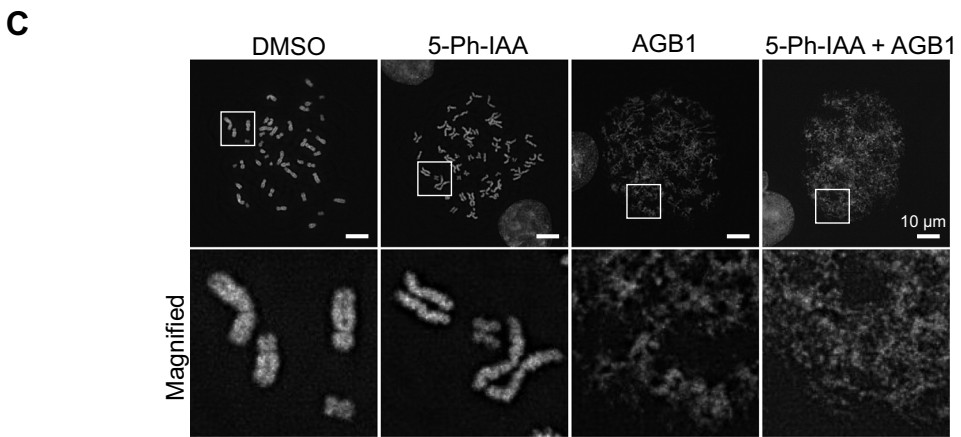

◄ **Figure 3. Independent and simultaneous depletion of RAD21-mAID-mClover (RAD21-mACl) and SMC2-BromoTag-mCherry2 (SMC2-BTCh).**

(A) Schematic illustration showing degradation of RAD21-mACl and SMC2-BTCh in the cohesin and condensin complexes, respectively. (B) Top, density plots of the cells treated with 1 µM 5-Ph-IAA, 0.5 µM AGB1 and both for 2 h. The X and Y axes are the signal intensity of RAD21-mACl and SMC2-BTCh, respectively. Quadrant gates (Q1–Q4) were set manually to include 99.5% of the single cells in the Q4 area using a negative control (DMSO-treated WT cells). Bottom, the density plot data were quantified to make this graph. Data were presented as mean ± SD of three technical replicates ($n = 3$). The signal intensities in each condition were normalized with DMSO-treated control cells. (C) Representative DAPI-stained chromosomes from RAD21-mACl/SMC2-BTCh cells after treating them with the indicated ligand for 3 h. The scale bars indicate 10 µm. Source data are available online for this figure.

subsequently, the GFP reporter level was monitored using a flow cytometer. In the case of AID2, efficient depletion was achieved with the lowest concentration (DC50 = 0.2 nM 5-Ph-IAA). There are two inducing ligands for the dTAG system, dTAG-13 and dTAGv-1, which recruit CRL4–CRBN and CRL2–VHL E3 ligase, respectively (Appendix Fig. S1A) (Nabet et al, 2020; Nabet et al, 2018). We found that dTAG-13 did not efficiently deplete the GFP reporter in HCT116 cells within 4 h. On the other hand, dTAGv-1 worked as expected and achieved almost complete depletion at 50 nM (DC50 = 0.9 nM). The reason why dTAG-13 did not work efficiently in HCT116 cells might be because the expression levels of CRL4–CRBN, which recognizes dTAG in the presence of dTAG-13, were lower than those of the other E3 ubiquitin ligases required by 5-Ph-IAA, dTAGv-1 and AGB1 (Appendix Fig. S1B) (Bekker-Jensen et al, 2017). AGB1 for BromoTag also induced depletion of the GFP reporter, albeit slightly less efficiently (DC50 = 3.7 nM). To study whether the trends we found were also true in another cell line, we generated a similar reporter cell line in a normal human cell line, hTERT-RPE1, and found a similar trend, although dTAG-13 was more effective in this cell line (Appendix Fig. S2B). Unlike the AID2 system, PROTAC dTAGs and AGB1 showed the hook effect at high doses of ligands, a characteristic feature of heterobifunctional degraders (Fig. 2B; Appendix Fig. S2B).

Next, we investigated depletion kinetics by treating the HCT116 reporter cells with 100 nM ligand and then took time-course samples (Fig. 2C). We found that T1/2 was 67, 82 and 62 min when treated with 5-Ph-IAA, dTAGv-1 and AGB1, respectively, indicating AID2 and BromoTag comparably induced slightly faster depletion than dTAG with dTAGv-1. We found that target depletion with dTAG-13 was slow in HCT116 cells (Fig. 2C dTAG + 100 nM dTAG-13), and it took 12 h for the GFP reporter level to decrease to less than 10% (Appendix Fig. S2C). We also studied the depletion kinetics in the hTERT-RPE1 reporter cells and found T1/2 of AID2, dTAG (with dTAGv-1), and BromoTag were 13, 62, and 43 min, respectively (Appendix Fig. S2D). These results indicated that AID2 and BromoTag depleted the GFP reporter quicker than dTAG in both HCT116 and hTERT-RPE1 cells.

An ideal inducible degron system should operate reversibly. To investigate whether the GFP reporter can be re-expressed after depletion, we initially treated the HCT116 GFP reporter cells with 100 nM 5-Ph-IAA, dTAGv-1 or AGB1 for 4 h to induce GFP reporter depletion. Subsequently, the cells were washed and incubated in fresh medium without each ligand to monitor the re-expression of the GFP reporter (Fig. 2D). As previously reported, the cells treated with 5-Ph-IAA recovered GFP reporter expression over time, with 8 h required to achieve 50% recovery (Fig. 2D, AID2) (Yesbolatova et al, 2020). Unexpectedly, the cells treated with dTAGv-1 did not show any recovery until 24 h (Fig. 2D, dTAG). The cells treated with AGB1 recovered up to about 30% at

6 h, but no further recovery was observed (Fig. 2D, BromoTag). We found similar results in expression recovery in hTERT-RPE1 cells (Appendix Fig. S3A). To investigate the recovery after target depletion further, we treated the HCT116 reporter cells with the DC50 concentration of 5-Ph-IAA, dTAGv-1, or AGB1 identified in Fig. 2B and found that the reporter level decreased to approximately 30% after 12 h (Appendix Fig. S3B, 12 h). We then washed the cells and monitored reporter re-expression. We found that the reporter level was almost completely recovered (94.2 ± 4.6%) after 24 h in cells treated with 5-Ph-IAA (Appendix Fig. S3B). In contrast, the cells treated with dTAGv-1 or AGB1 recovered to 69.7 ± 8.3% and 78.9 ± 12.5%, respectively. These results indicate that PROTAC-based degrons exhibited poor recovery after target depletion, suggesting that dTAGv-1 and AGB1 continued to bind CRL2–VHL E3 ligase even after medium exchange, maintaining target degradation, consistent with their catalytic substoichiometric mode of action at low concentrations. Considering the results of ligand concentration (Fig. 2B; Appendix Fig. S2B), depletion kinetics (Fig. 2C; Appendix Fig. S2D), and expression recovery (Fig. 2D; Appendix Fig. S3), we decided to combine AID2 and BromoTag in the following experiments.

## Independent and simultaneous depletion of two proteins

Because we found that AID2 and BromoTag achieved rapid depletion of the GFP reporter (Fig. 2C), we asked whether two proteins can be independently depleted with AID2 and BromoTag. For this purpose, we chose two structural maintenance-of-chromosome (SMC) complexes, cohesin and condensin, which are involved in sister-chromatid cohesion and mitotic chromosome formation, respectively (Jeppsson et al, 2014). We fused mAID-mClover to the C-terminus of the endogenous RAD21 subunit of cohesin as previously reported (Appendix Fig. S4A) (Natsume et al, 2016). Similarly, we introduced BromoTag-mCherry2 to the endogenous SMC2 subunit of condensin. The established HCT116 cell line expressed both RAD21-mAID-mClover (RAD21-mACl) and SMC2-BromoTag-mCherry2 (SMC2-BTCh), which are induced for rapid degradation in the presence of 5-Ph-IAA and AGB1, respectively (Fig. 3A). Initially, we confirmed the expression of RAD21-mACl and SMC2-BTCh by fluorescence microscopy (Appendix Fig. S4B, control).

Before depleting RAD21-mACl and SMC2-BTCh in this cell line, we optimized inducing concentrations of 5-Ph-IAA and AGB1. We cultured the parental HCT116 WT cells in the presence of 5-Ph-IAA or AGB1 for 7 days (Appendix Fig. S4C). We found that 1 µM 5-Ph-IAA did not affect cellular proliferation and colony formation as previously reported (Yesbolatova et al, 2020). On the other hand, the cells grown in 1 µM AGB1 formed smaller colonies, suggesting that AGB1 affected cellular proliferation. The cells grown in 0.5 µM AGB1 formed colonies as in the control

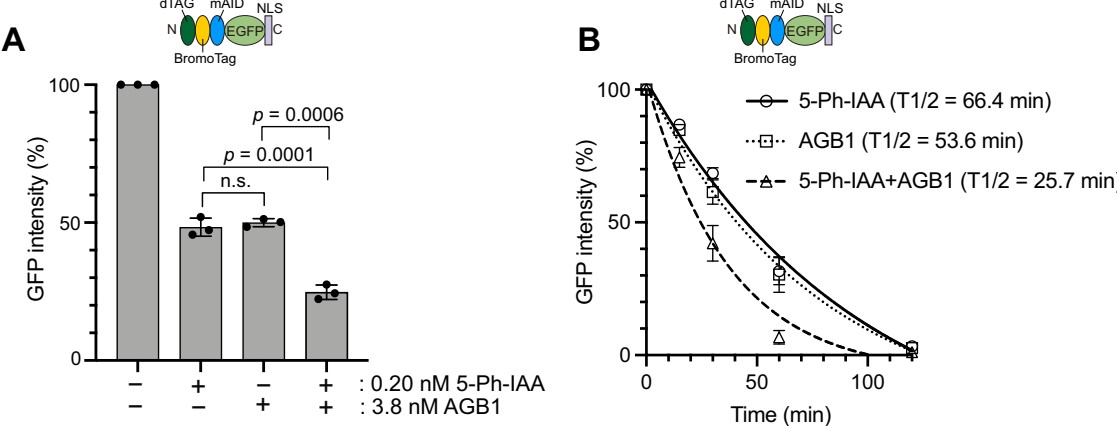

**Figure 4. Enhancing reporter depletion by a double-degron system with AID2 and BromoTag.**

(A) The reporter cells used in Figs. 2 and 3 were treated with 0.2 nM 5-Ph-IAA and/or 3.8 nM AGB1 for 4 h. GFP intensity was monitored taking the mock-treated cells as 100%. Data were presented as mean ± SD. Each dot represents a technical replicate ($n = 3$). (B) Depletion kinetics of the reporter depletion in cells treated with 1 μM 5-Ph-IAA, 0.5 μM AGB1, or both. The GFP intensity was monitored at 0, 15, 30, 60, 120, and 240 min taking the mock-treated cells as 100%. Data were presented as mean ± SD of three technical replicates ($n = 3$). The data were fitted with one-phase decay. Source data are available online for this figure.

DMSO-added culture. Therefore, we decided to employ 1 μM 5-Ph-IAA and 0.5 μM AGB1 for the following experiments.

We treated the cells with 5-Ph-IAA or AGB1 independently, or together. After treating the cells, the expression level of RAD21-mACl and SMC2-BTCh was monitored by flow cytometry and microscopy (Fig. 3B; Appendix Fig. S4B). When treated with 5-Ph-IAA or AGB1, each target protein was depleted without affecting the other protein. The addition of both 5-Ph-IAA and AGB1 combined resulted in the depletion of both RAD21-mACl and SMC2-BTCh. These data clearly show that two endogenous proteins were independently or simultaneously depleted with the orthogonal AID2 and BromoTag systems.

Subsequently, we investigated phenotypic defects in chromosomal structures after the depletion of RAD21-mACl and SMC2-BTCh. For this purpose, we looked at mitotic chromosomes after RAD21-mACl and SMC2-BTCh depletion (Fig. 3C). Chromosomes in the 5-Ph-IAA-treated cells (RAD21-depleted cells) showed loss of sister-chromatid cohesion (Fig. 3C, 5-Ph-IAA) (Sonoda et al, 2001; Yesbolatova et al, 2020). Chromosomes in AGB1-treated cells (SMC2-depleted cells) showed the rod-like structure of mitotic chromosomes was highly disorganized, as previously reported for loss of condensin (Fig. 3C, AGB1) (Boteva et al, 2020; Takagi et al, 2018). Furthermore, chromosomes in the 5-Ph-IAA- and AGB1-treated cells (RAD21- and SMC2-depleted cells) were even more disorganized than those treated with AGB1 alone, possibly because of an additive effect of cohesion and structural losses (Fig. 3C, 5-Ph-IAA + AGB1). These results demonstrate that two proteins can be independently and simultaneously depleted in a single cell by utilizing AID2 and BromoTag, allowing dissection of the functional consequences of knocking down two proteins individually or concurrently.

## Enhancing target protein depletion by an AID2-BromoTag double-degron system

As we presented in Fig. 1, there are cases in which target depletion by AID2 does not result in a strong phenotypic defect. We hypothesized

that a double-degron tag composed of mAID and BromoTag in tandem would enhance target protein depletion, thus causing stronger phenotypic defects. To test this idea, we investigated whether the GFP reporter used in Fig. 2 can be depleted better by treating the reporter cells with both 5-Ph-IAA and AGB1. We initially identified that treatment with 0.2 nM 5-Ph-IAA or 3.8 nM AGB1 caused about 50% reduction in expression of the GFP reporter (48.3% ± 3.3 or 50% ± 1.5, respectively) (Fig. 4A, bars 2 and 3). Subsequently, we treated the GFP reporter cells with both 0.2 nM 5-Ph-IAA and 3.8 nM AGB1 and found that the GFP expression level was 24.7% ± 2.6, indicating an additive degradation by AID2 and BromoTag (Fig. 4A, bar 4).

We were encouraged by the additive effect in the expression level and asked whether depletion kinetics can also be accelerated. To answer this question, we treated the GFP reporter cells with 1 μM 5-Ph-IAA, or 0.5 μM AGB, or both, and then took time-course samples. Figure 4B clearly shows that depletion kinetics was accelerated by treating both ligands (T1/2 = 25.7 min). Single ligand treatment achieved comparable depletion kinetics (5-Ph-IAA T1/2 = 66.4 min, AGB1 T1/2 = 53.6 min; not statistically significant by paired *t*-test). These results demonstrated that the AID2-BromoTag double-degron system induced enhanced degradation and accelerated depletion kinetics compared with the case of the single degron.

## ORC1 and CDC6 depletion by the AID2-BromoTag double-degron system

Considering the results with the GFP reporter shown above, we then investigated whether AID2-BromoTag will give better depletion and phenotypic defects for ORC1. We fused mAID-BromoTag to the N-terminus of endogenous ORC1(mAB-ORC1). The mAB-ORC1 cells were treated with 5-Ph-IAA, AGB1, or both for 2 h (Fig. 5A). Compared with the mAB-ORC1 expression level treated with 5-Ph-IAA or AGB1 (16.6 or 7.7%, respectively), expression was further reduced when treated with both (3.9%), similar to the case observed with the GFP reporter in Fig. 4.

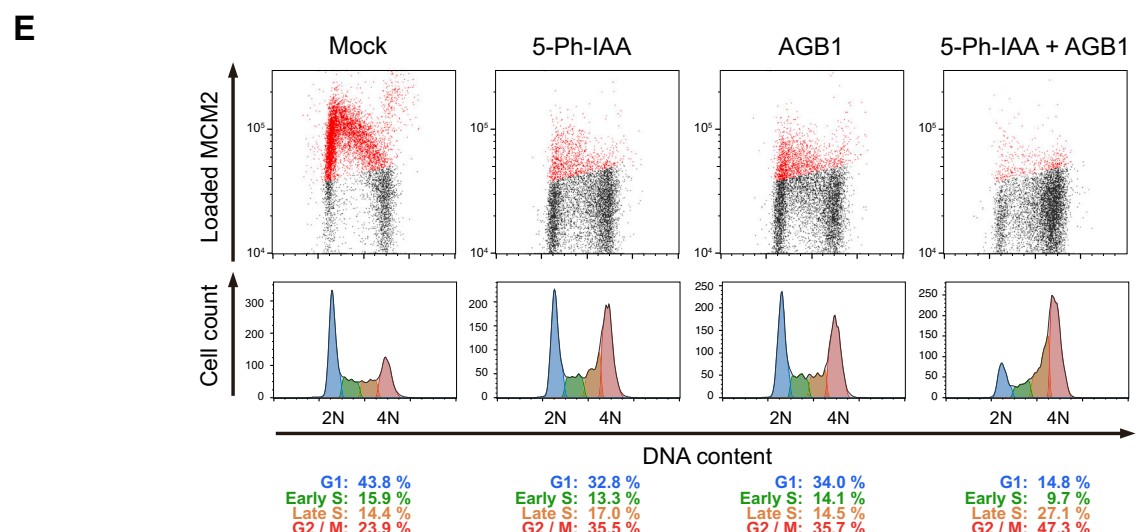

Figure 5.  Double-degron with mAID and BromoTag enhances ORC1 depletion and confers profound defects in DNA replication.

(A) The parental HCT116 wild-type (WT) and mAID-BromoTag-ORC1 (mAB-ORC1) cells were treated with 1 µM 5-Ph-IAA, 0.5 µM AGB1 or both for 2 h. Proteins were detected by anti-ORC1 and -tubulin antibodies. Relative ORC1 levels taking the DMSO-treated control as 100% are shown under each blot. Each data point was normalized with the corresponding tubulin loading control. The asterisk indicates the HygR-P2A-mAID-BromoTag-ORC1 protein before self-cleavage at the P2A site. (B) Depletion kinetics of mAB-ORC1 in cells treated with 1 µM 5-Ph-IAA and/or 0.5 µM AGB1. Samples were taken at the indicated time points. (C) The blot data in panel B were quantified, taking the 0 min sample as 100%. Each data point was normalized with the corresponding tubulin loading control. (D) Colony formation of the parental HCT116 WT and mAB-ORC1 cells. Cells were cultured in the presence or absence of 1 µM 5-Ph-IAA and/or 0.5 µM AGB1 for 7 days. Colonies were stained with crystal violet. (E) (Upper panels) Levels of chromatin-loaded MCM2 and DNA in mAB-ORC1 treated with 1 µM 5-Ph-IAA and/or 0.5 µM AGB1 for 24 h. The MCM2-positive cells are shown in red. (Lower panels) Cell count histogram for the same samples. The percentages of each cell cycle are shown below. (F) Levels of chromatin-loaded MCM2 in mAB-ORC1 cells treated with the indicated ligand. Data were presented as mean ± SD. Each dot represents a technical replicate ($n = 3$). Cells were synchronized in M phase with 50 ng/mL nocodazole for 14 h and released into fresh media containing ligand. Cells were treated with 1 µM 5-Ph-IAA and/or 0.5 µM AGB1 2 h prior to nocodazole release. Samples were taken at 4 h after release when cells were in G1. Statistical analysis was performed with Tukey's multiple comparison test. The p values are 1.53e-05 for 5-Ph-IAA vs AGB1, 3.43e-07 for AGB1vs 5-Ph-IAA + AGB1, and 4.77E-04 for 5-Ph-IAA vs 5-Ph-IAA + AGB1, respectively. Source data are available online for this figure.

Subsequently, we investigated the depletion kinetics of mAB-ORC1 treated with 5-Ph-IAA, AGB1, or both, and then took time-course samples (Fig. 5B,C). At every time point, the cells treated with both ligands showed better depletion of mAB-ORC1, indicating that mAB-ORC1 depletion was enhanced and accelerated by treating with 5-Ph-IAA and AGB1 together.

Next, we studied the viability of the mAB-ORC1 cells by culturing them in the presence of 5-Ph-IAA, AGB1 or both (Fig. 5D). Cells grown with 5-Ph-IAA made small colonies as observed with mAID-ORC1 cells in Fig. 1C. Similarly, cells grown with AGB1 formed small colonies, although the colonies were slightly bigger than those of the 5-Ph-IAA-treated cells. In contrast, the cells with both 5-Ph-IAA and AGB1 did not form colonies, showing that mAB-ORC1 depletion caused lethality.

ORC1 plays an essential role in loading MCM2–7 to chromosomal DNA in late M to G1 phases in budding yeast (Klemm and Bell, 2001). We investigated whether mAB-ORC1 depletion would affect MCM2–7 loading in human cells by looking at chromatin-bound MCM2 (Fig. 5E) (Haland et al, 2015). The mAB-ORC1 cells were cultured with 5-Ph-IAA, AGB1, or both for 24 h. Subsequently, chromatin-bound MCM2 was stained after extraction of chromatin-unbound MCM2. The level of chromatin-bound MCM2 shown in red was reduced in the cells treated with 5-Ph-IAA or AGB1 (Fig. 5E). However, chromatin-bound MCM2 was even more reduced in the cells treated with both 5-Ph-IAA and AGB1. We confirmed this notion by synchronizing cells and quantified the levels of chromatin-bound MCM2 in the G1 phase (Fig. 5F). Furthermore, the proportion of cells arrested in late S to G2 phases was the highest with 5-Ph-IAA and AGB1, suggesting these cells were experiencing strong defects in DNA replication with small amounts of chromatin-loaded MCM2–7 (Fig. 5E, 5-Ph-IAA + AGB1). Consistent with this interpretation, we observed 53BP1 foci showing that DNA damage was highly accumulated in the cells treated with both 5-Ph-IAA and AGB1 for 43 h, as compared with those treated with either 5-Ph-IAA or AGB1 (Appendix Fig. S5A,B). Considering these results, we concluded that ORC1 is pivotal for loading MCM2–7 onto chromatin DNA and, thus, for driving human cell proliferation.

Next, we wished to test the AID2-BromoTag double-degron system with another DNA replication factor. It has been reported that the original AID system was employed for depleting CDC6, which collaborates with ORC1–6 for loading MCM2–7 (Lemmens et al, 2018). However, the depletion was insufficient to observe defects in DNA replication. Therefore, we fused mAID-BromoTag to the N-terminus of the endogenous CDC6 protein (mAB-CDC6)

and carried out similar experiments to those done for mAB-ORC1 (Appendix Fig. S6). When we treated mAB-CDC6 cells with 5-Ph-IAA, AGB1, or both for 2 h, we observed an additive depletion resulting in reduced expression of mAB-CDC6 down to 9% (Appendix Fig. S6A). Depletion of mAB-CDC6 was accelerated by treating with both 5-Ph-IAA and AGB1 (Appendix Fig. S6B,C). Importantly, treating with both 5-Ph-IAA and AGB1 was lethal to the mAB-CDC6 cells (Appendix Fig. S6D). In contrast, a single treatment with 5-Ph-IAA or AGB1 led to slow growth. Furthermore, treating with both 5-Ph-IAA and AGB1 induced strong defects in the loading of MCM2–7, arresting cells in late S to G2 phases with 53BP1 foci accumulation (Appendix Figs. S5 and S6E,F). Taken together, we concluded that the double-degron system with AID2 and BromoTag is more effective than the respective single-degron at enhancing target depletion and inducing pronounced phenotypic defects. We also concluded that both ORC1 and CDC6 are pivotal for MCM loading, the essential process prerequisite for DNA replication.

## Tandem mAID tags reduce target expression even though they induce strong phenotypic defects

Because the double-degron system with AID2 and BromoTag enhanced target depletion, we wondered whether utilizing a tandem mAID tag would also enhance target depletion, as previously reported in yeast (Kubota et al, 2013; Nishimura and Kanemaki, 2014). We fused one, two, or three copies of mAID (mAID, 2mAID, or 3mAID, respectively) to the N-terminus of ORC1 and compared them with each other (Fig. 6). We found that the expression level of 2mAID- and 3mAID-ORC1 was significantly reduced compared with that of mAID-ORC1 (Fig. 6A, – 5-Ph-IAA). This instability was also observed in cells expressing 2mAID-CDC6 (Appendix Fig. S7A). The reduced expression of 2mAID-ORC1 is independent of leaky degradation by OsTIR1(F74G) (Appendix Fig. S7B). We found that 2mAID did not affect the protein half-life, but 3mAID conferred a shorter half-life, suggesting that 3mAID caused instability in the fusion protein (Appendix Fig. S7C). We also noted that the mRNA levels of 2mAID-ORC1 was significantly reduced (Appendix Fig. S7D), suggesting that the insertion of the 2mAID cassette into the N-terminus coding region of the ORC1 gene affected its expression. Even though the expression of 2mAID- and 3mAID-ORC1 was reduced, the cells proliferated normally in the absence of 5-Ph-IAA (Fig. 6B,+DMSO). Upon the addition of 5-Ph-IAA, 2mAID- and 3mAID-ORC1 were effectively depleted (Fig. 6A) and showed a lethal phenotype (Fig. 6B,+5-Ph-IAA). As expected, the treated cells were defective in loading MCM2–7

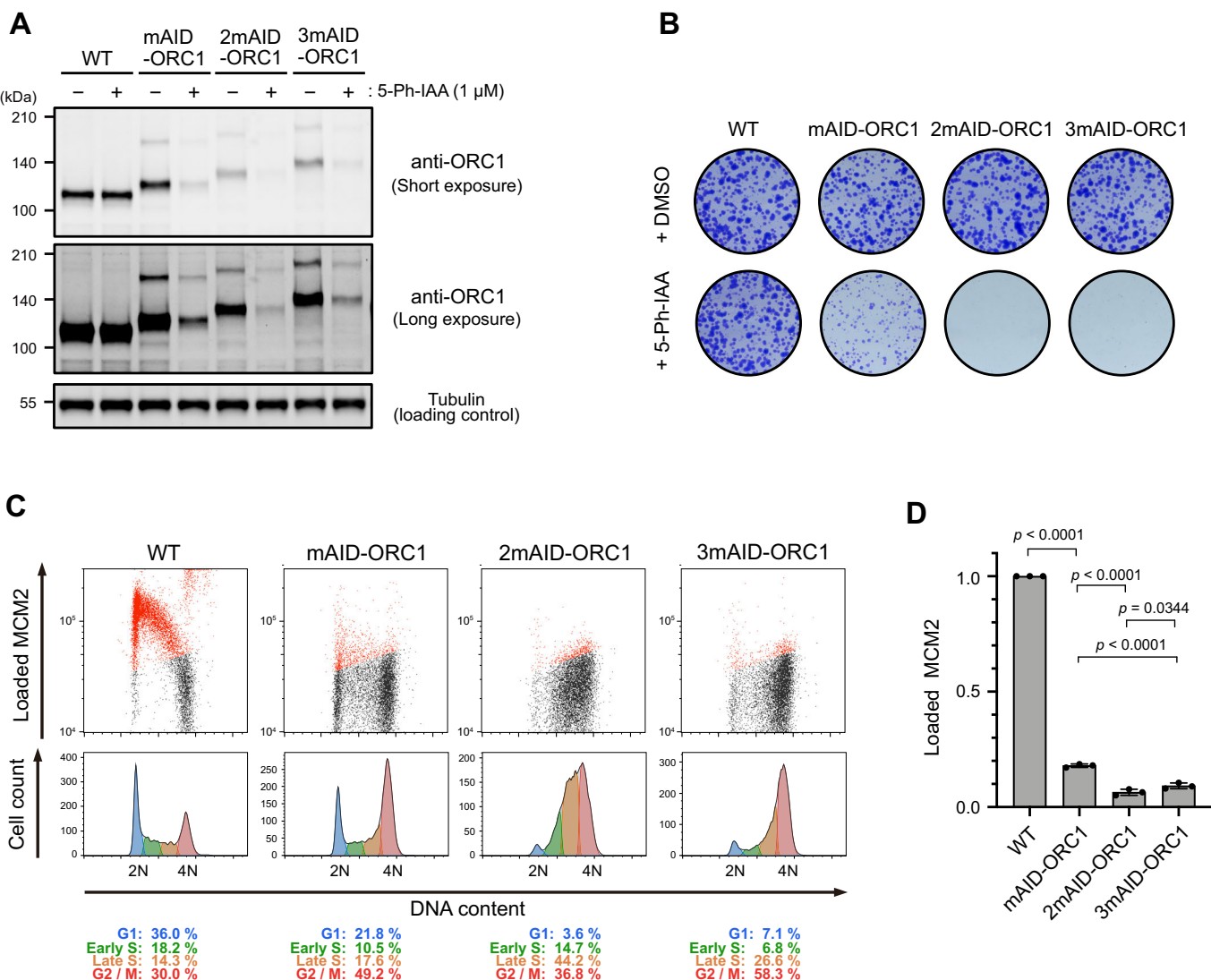

**Figure 6. ORC1 depletion and phenotypic defects of mAID-, 2mAID- and 3mAID-ORC1 cells.**

(A) ORC1 level in cells treated with 1 μM 5-Ph-IAA for 2 h. (B) Colony formation of the indicated cells. Cells were cultured in the presence or absence of 1 μM 5-Ph-IAA for 7 days. Colonies were stained with crystal violet. (C) (Upper panels) Level of chromatin-loaded MCM2 and DNA in the indicated cells. Cells were cultured with 1 μM 5-Ph-IAA for 24 h. MCM2-positive cells are shown in red. (Lower panels) Cell count histogram for the same samples. The percentages of each cell cycle are shown below. (D) Levels of chromatin-loaded MCM2 in the G1 phase of indicated cell lines treated with 5-Ph-IAA. Data were presented as mean ± SD. Each dot represents a technical replicate ($n = 3$). Cells were synchronized in the M phase with 50 ng/mL nocodazole for 14 h and released into fresh media containing 5-Ph-IAA. Cells were treated with 1 μM 5-Ph-IAA 2 h prior to nocodazole release. Samples were taken at 4 h after release when cells were in G1. Statistical analysis was performed with Tukey's multiple comparison test. The $p$ values are 5.16e-14 for WT vs mAID-ORC1, 2.54e-06 for mAID-ORC1 vs 2mAID-ORC1, 3.44e-02 for 2mAID-ORC1 vs 3mAID-ORC1, and 2.04e-05 for mAID-ORC1 vs 3mAID-ORC1, respectively. Source data are available online for this figure.

onto chromatin and arrested in the late-S to G2 phase (Fig. 6C,D). Therefore, we concluded that 2mAID and 3mAID allow for stronger phenotypic defects, with the caveat that they might reduce the expression levels of the fusion protein.

## Complete inhibition of DNA replication leads to uncoupling DNA replication from the cell cycle control

Up until this point, we had succeeded in depleting ORC1 and CDC6 using the AID2-BromoTag double-degron system, and demonstrated their importance for MCM2–7 loading and cellular proliferation (Fig. 5 and Appendix Fig. S6). However, mAB-ORC1 and mAB-CDC6 cells replicated genomic DNA mostly and arrested around 4 N (Fig. 5E; Appendix Fig. S6E). The same was true when we used 2mAID-ORC1 (Fig. 6C). These results suggest two intriguing implications. First, human HCT116 cells do not have the licensing checkpoint, monitoring the amount of chromatin-bound MCM2–7 in the G1 phase because they carried out DNA replication with reduced chromatin-bound MCM2–7, consistent with our previous report (Saito et al, 2022). Second, the results suggest the ORC1 or CDC6 depletion might have been still not enough to completely suppress MCM loading and DNA replication.

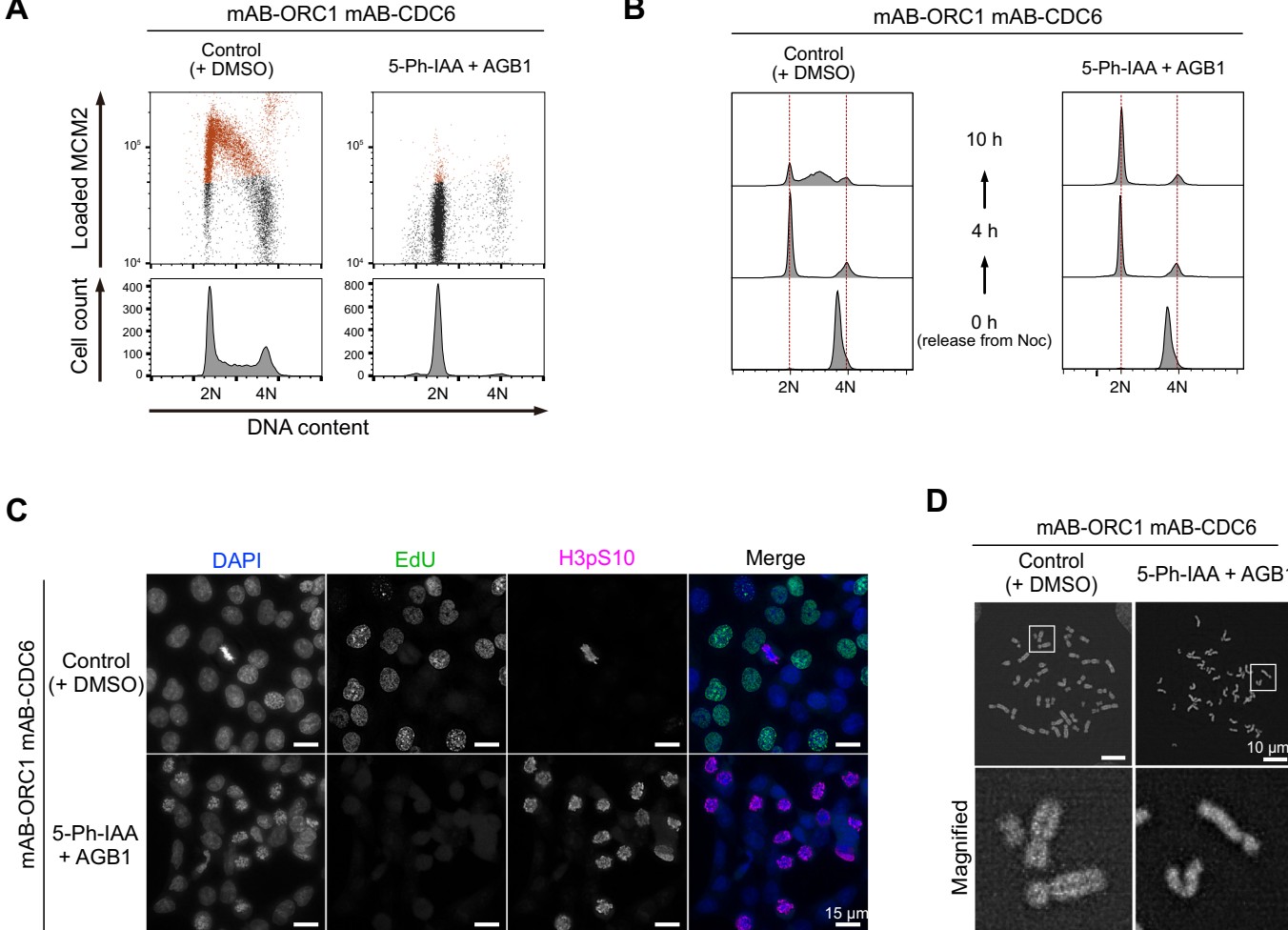

**Figure 7. Complete inhibition of DNA replication in the mAB-ORC1 mAB-CDC6 cells leads to premature mitosis.**

(A) (Upper panels) Levels of chromatin-loaded MCM2 and DNA in the mAB-ORC1 mAB-CDC6 cells treated with 1 μM 5-Ph-IAA for 24 h. The MCM2-positive cells are shown in red. (Lower panels) Cell count histogram for the same samples. (B) Cell cycle progression of the mAB-ORC1 mAB-CDC6 cells. Cells were synchronized with 50 ng/mL nocodazole for 14 h and then released into fresh media containing 1 μM 5-Ph-IAA and 0.5 μM AGB1. Cells were treated with 5-Ph-IAA and AGB1 2 h prior to nocodazole release. Samples were taken at the indicated time points. (C) Fluorescent microscopic images of mAB-ORC1 mAB-CDC6 cells cultured with or without 1 μM 5-Ph-IAA and 0.5 μM AGB1 for 24 h. Cells were treated with 10 μM EdU for 30 min before fixation. Cells under mitosis were stained with an anti-histone H3 p-Ser10 antibody. The scale bars indicate 15 μm. (D) DAPI-stained spread chromosomes after treating the mAB-ORC1 mAB-CDC6 cells with or without 1 μM 5-Ph-IAA and 0.5 μM AGB1 for 24 h. The scale bars indicate 10 μm. Source data are available online for this figure.

To clarify the second issue, we combined RNA knockdown with siRNA and protein knockdown with AID2-BromoTag. Initially, we transfected siRNA to the mAB-ORC1 and -CDC6 cells 12 h prior to the addition of 5-Ph-IAA and AGB1 (Appendix Fig. S8A). Subsequently, we cultured the cells for 24 h in the presence of the ligands and confirmed that the expression levels of the mAB-ORC1 and -CDC6 proteins reached close to the detection limit by Western blotting (Appendix Fig. S8B). When siRNA and double-degron were combined, both the mAB-ORC1 and -CDC6 cells showed enhanced defects in the progression of the S phase (Appendix Fig. S8C,D). Therefore, we interpreted that MCM-loading still occurred in the mAB-ORC1 or -CDC6 depleted cells by AID2-BromoTag using a low level of the remaining protein (Fig. 5E; Appendix Fig. S6E). Even though mAB-ORC1 was depleted close to the detection limit using siRNA and AID2-BromoTag, the cells carried

out DNA replication to some extent (Appendix Fig. S8C). This data suggests two possibilities, which are hard to clarify: there was still mAB-ORC1 existing even after the combinational depletion, or MCM-loading occurred to some extent without ORC1. In contrast, it is likely that CDC6 is absolutely essential for MCM-loading (Appendix Fig. S8D).

To suppress licensing further, we generated a double-mutant cell line for ORC1 and CDC6: mAB-ORC1 mAB-CDC6. We found that MCM loading was completely suppressed, arresting the cells at 2N when ORC1 and CDC6 were simultaneously depleted (Fig. 7A; Appendix Fig. S9A). Furthermore, we observed that DNA replication was completely suppressed in a synchronous culture released from nocodazole arrest (Fig. 7B). Interestingly, we found that the simultaneous depletion of ORC1 and CDC6 led to premature mitosis without DNA replication (Fig. 7C,D). In the

rounded cells, histone H3 Ser10 was highly phosphorylated, showing they are in mitosis (Fig. 7C). In these cells, condensed chromosomes did not show sister chromatids; instead, they showed single-chromatids (Fig. 7D). Furthermore, these single-chromatid chromosomes failed to align at the metaphase plate, even though spindle formation normally occurred (Appendix Fig. S9B). This failure was likely due to the inability to establish a centromeric bipolar attachment to microtubules. We concluded that ORC1 and CDC6 co-depletion by the AID2-BromoTag double-degron system completely suppresses DNA replication, leading to uncoupling DNA replication from the cell cycle control (Appendix Fig. S9C).

## Discussion

In this paper, we initially compared AID2, dTAG, and BromoTag to learn their similarities and differences (Fig. 2; Appendix Figs. S2 and S3). While 5-Ph-IAA, dTAGv-1, and AGB1 effectively depleted the GFP reporter in HCT116 and hTERT-RPE1, dTAG-13 was not as effective, especially in HCT116, as reported in other cell lines by others (Li et al, 2019; Olsen et al, 2022). This is likely due to a relatively low expression of the CRL4–CRBN components in HCT116 cells (Appendix Fig. S1B). The expression levels of the responsible E3 ligase likely affected depletion efficiency and kinetics because 5-Ph-IAA induced more highly profound reporter depletion in hTERT-RPE1 cells than in HCT116 cells (Fig. 2B,C; Appendix Fig. S2B,D). Indeed, OsTIR1(F74G) in hTERT-RPE1 is overexpressed by random integration, while it is moderately expressed from the AAVS1 locus in HCT116. Additionally, we found that the reversible recovery after depletion with dTAG and BromoTag was poor compared with AID2 (Fig. 2D; Appendix Fig. S3), which is consistent with another report (Bondeson et al, 2022). We hypothesize that target degradation continued because the substoichiometric activity of PROTACs dTAGv-1 and AGB1 bound to the CRL2–VHL E3 ligase can continue to be effective even after washing the compounds out from the culture medium. It will be important to test how generally applicable such features of these potent degron-based PROTAC degraders will be across other cell types/conditions and ultimately in vivo. One point that should be noted is that we focused on targeting nuclear proteins in this study. While AID2, dTAG, and BromoTag are reported to degrade proteins in the cytoplasm (Bond et al, 2021; Nabet et al, 2020; Yesbolatova et al, 2020), the degradation efficiency and depletion kinetics in the cytoplasm might be different compared to the nucleus due to the localization and activity of the responsible E3 ligase.

Controlling the expression of two proteins is useful when studying their epistatic relationship and functional redundancy. We demonstrated that two proteins can be independently as well as simultaneously depleted by utilizing AID2 and BromoTag. As a proof-of-concept experiment, we depleted cohesin RAD21 and condensin SMC2 with AID2 and BromoTag, respectively (Fig. 3). As expected, RAD21 or SMC2 depletion caused defects in sister-chromatid cohesion and in forming rod-like chromosomes, respectively (Fig. 3C) (Boteva et al, 2020; Sonoda et al, 2001; Takagi et al, 2018; Yesbolatova et al, 2020). Interestingly, the mitotic chromosomes depleted of both RAD21 and SMC2 were highly disorganized compared with those depleted of only SMC2. Our interpretation was that this chromosomal phenotype was an additive effect of sister-chromatid cohesion and chromosomal axis formation. It will be interesting to reveal what actually happens to the mitotic chromosomes in RAD21- and SMC2-depleted cells by Hi-C or super-resolution imaging approaches.

We succeeded in developing a tandem-degron system with mAID-BromoTag to enhance target depletion and accelerate depletion kinetics (Figs. 4 and 5; Appendix Fig. S6). Empirically, we have achieved a near-knockout phenotype with AID2 for many replication proteins (Klein et al, 2021; Lim et al, 2023; Liu et al, 2023; Saito et al, 2022). However, we have found that there were some cases, such as ORC1 and CDC6, in which we could not deplete them sufficiently to study phenotypic defects. In these instances, the tandem-degron system proved to be advantageous. Moreover, each degron system shows different degradation efficiency depending on the protein of interest (POI) (Bondeson et al, 2022). Therefore, combining two degron systems is expected to effectively deplete more POIs than a single degron. We also found that tandem mAID tags (2mAID and 3mAID) were also effective for ORC1, causing strong defects in proliferation and MCM loading to chromatin DNA (Fig. 6). However, these tandem tags reduced the expression level of ORC1 independent of the presence or absence of OsTIR1(F74G) (Fig. 6A; Appendix Fig. S7). We previously reported that 3mAID was more effective than 2mAID in budding yeast (Kubota et al, 2013; Nishimura and Kanemaki, 2014). However, ORC1 was more effectively depleted with 2mAID than with 3mAID in human cells (Fig. 6A). Therefore, the tandem 2mAID tag can be used if the reduced expression level of a POI without 5-Ph-IAA does not cause a problem. Because the mRNA level was reduced in cells expressing 2mAID-ORC1 (Appendix Fig. S7D), it will be interesting to test whether 2mAID knock-in will affect the mRNA levels of other genes.

We demonstrated that MCM2-depleted cells by AID2 showed a lethal phenotype (Fig. 1B,C). On the other hand, we found that cells survived with a small amount of ORC1 and CDC6, and we needed to further deplete them by utilizing a mAID-BromoTag double degron to observe lethality (Figs. 1B,C and 5A,D; Appendix Fig. S6A,D). These results clearly indicate that the level of depletion required to cause phenotypic defects is variable even though MCM2, ORC1, and CDC6 are involved in DNA replication. The ORC1–6 complex and CDC6 are all required for loading the MCM2–7 complex to chromatin DNA (Costa and Diffley, 2022; Li et al, 2023), and a single ORC1–6 and CDC6 can potentially load multiple MCM double hexamers. Conversely, the loaded MCM double hexamers are converted to form the replicative CMG helicases for making the replication forks, and excess MCM2–7 is required for robust DNA replication (Ge et al, 2007; Ibarra et al, 2008; Peycheva et al, 2022). Their differential roles in DNA replication likely define the minimal amount needed for cell survival. In this study, we succeeded in effectively depleting ORC1 and CDC6 utilizing the double-degron system, causing defects in MCM loading and a lethal phenotype (Fig. 5; Appendix Fig. S6). Interestingly, after ORC1 or CDC6 depletion, cells entered S phase with a small amount of MCM2–7 on chromatin (Fig. 5E; Appendix Fig. S6E) and arrested in late-S and G2 phases, accumulating DNA damage (Appendix Fig. S5). Our interpretation is that HCT116 cells do not have the licensing checkpoint even though they are a p53-positive cell line, as we previously reported (McIntosh and Blow, 2012; Saito et al, 2022). These cells did not sense the lack of chromatin-bound MCM2–7 in a single cell cycle, entered S phase

and then activated the DNA damage checkpoint. Because the double-degron system depletes the target very fast, it can be used for studying whether the licensing checkpoint exists in other cell lines. Regarding the conflicting reports on whether ORC1 is essential for DNA replication (Chou et al, 2021; Shibata et al, 2016), our data support the conclusion that ORC1 is pivotal for loading MCM2–7 to drive cell proliferation (Figs. 5 and 6; Appendix Fig. S8). However, we do not exclude the possibility that MCM-loading might occur to some extent without ORC1, even though the amount of which is not enough for cells to proliferate (Appendix Fig. S8C). Furthermore, we succeeded in demonstrating DNA replication was completely suppressed in cells depleted of ORC1 and CDC6 simultaneously, leading the cells to premature mitosis without DNA replication (Fig. 7). Therefore, we concluded DNA replication was artificially uncoupled from the cell cycle control (Appendix Fig. S9C), as shown by another report (Lemmens et al, 2018). This result suggests that there is no cellular system monitoring the completion of DNA replication. Instead, cells monitor the presence of replication forks via the DNA damage checkpoint pathway (Fragkos et al, 2015).

In summary, we described a comparison of AID2, dTAG, and BromoTag, and our findings demonstrate that AID2 exhibited superior performance in terms of depletion efficiency, kinetics, and reversible expression recovery in HCT116 and hTERT-RPE1 cells. Furthermore, we created new degron tools to regulate two proteins and enhance target depletion. To show the power of our new tools, we demonstrated that both ORC1 and CDC6 play an essential role in loading MCM2–7 and showed complete suppression of DNA replication. We anticipate that the new degron tools presented in this study will be useful in various fields of cell biology and contribute to future studies.

# Methods

## Plasmids

All plasmids used in this study and other new tagging plasmids are listed in Table EV1. These plasmids and their sequence information are available from Addgene.

## Cell lines

All cell lines used in this study and their genotype are listed in Table EV2.

## Cell culture

HCT116 cells were cultured in McCoy's 5 A, supplemented with 10% FBS, 2 mM L-glutamine, 100 U/ml penicillin, and 100 μg/mL streptomycin at 37 °C in a 5% $CO_2$ incubator. To construct the GFP reporter cells expressing dTAG, BromoTag and mAID degrons tandemly, parental HCT116 cells expressing OsTIR1(F74G) were co-transfected with pMK459 (dTAG-BromoTag-mAID-EGFP-NLS) and pCMV-hyBase (PiggyBac transposase) using ViaFect Transfection reagent (Promega, #E498A) and Opti-MEM reduced serum medium (Thermo Fisher Scientific, #31985062) in a 12-well plate following the manufacturer's instruction (Yusa et al, 2011). Cells were selected with 100 μg/mL Hygromycin B Gold (Invivogen,

#ant-hg-5). Selected clones were isolated and GFP expression was confirmed by Western blotting and flow cytometry.

hTERT-RPE1 cells were cultured in D-MEM/Ham's F-12 with L-glutamine and Phenol Red, supplemented with 10% FBS, 100 U/ml penicillin, and 100 μg/mL streptomycin at 37 °C in a 5% $CO_2$ incubator. Initially, we established an hTERT-RPE1 stable cell line expressing OsTIR1(F74G) by introducing pMK444 (EF1-OsTIR1(F74G)) and pCMV-hyBase (Yusa et al, 2011). hTERT-RPE1 cells expressing OsTIR1(F74G) were co-transfected with pMK467 (dTAG-BromoTag-mAID-EGFP-NLS) and pCS-TP (TOL2 transposase) with the Neon system (Thermo Fisher Scientific) (Sumiyama et al, 2010). Cells were selected with 100 μg/mL Hygromycin B Gold. Selected clones were isolated and GFP expression was confirmed by Western blotting and flow cytometry.

All HCT116 cell lines expressing degron-fused protein(s) were generated following the protocol previously published (Saito and Kanemaki, 2021). Briefly, we constructed donor and CRISPR plasmids for tagging each gene. These CRISPR plasmids were designed to target MCM2 (5′-GAGGCCCTATGCCATCC/ATA-3′), ORC1 (5′-CCTTGTGGGGTAGTGTG/CCA-3′), CDC6 (5′-CATGCCTCAAACCCGAT/CCC-3′), RAD21 (5′-CCAAGGTTCCATATTAT/ATA-3′), and SMC2 (5′-ACCACCCAAAGGAGCAC/ATG-3′). After transecting both plasmids, cells were selected in the presence of an appropriate antibiotic. Single colonies were isolated and clones containing bi-allelic insertion at the target gene loci were selected by genomic PCR. The expression of degron-fused target protein was confirmed by Western blotting.

## Degrader compounds

5-Ph-IAA was synthesized as previously described (Yesbolatova et al, 2020). AGB1 for the initial studies was synthesized as previously reported (Bond et al, 2021), and later commercially obtained (Tocris #7686). dTAG-13 and dTAGv-1 were commercially obtained (Tocris #6605 and 6914, respectively).

## Colony formation assay

In a six-well plate, 1500 cells were seeded and cultured in the presence or absence of indicated ligands for 7 days. The culture medium was exchanged for fresh medium containing appropriate ligands on day 4. Cells were fixed and stained with crystal violet solution (6.0% Glutaraldehyde, 0.5% crystal violet).

## Flow cytometric analysis

Reporter cells in the HCT116 or hTERT-RPE1 background were seeded in a six-well plate, except for HCT116 RAD21-mACl/SMC2-BTCh cells, which were seeded in a 12-well plate. They were grown for 2 days to obtain 80–90% confluency and treated with ligands as indicated in the figure legends. For detecting GFP/mClover and mCherry signals after ligand treatment, cells were trypsinized and fixed in 4% methanol-free paraformaldehyde (PFA) at 4 °C for 20 min or overnight. Fixed cells were washed and resuspended in 1% BSA/PBS. Flow cytometric analysis was performed on a BD Accuri C6 (BD Biosciences) or a BD FACSCelesta (BD Biosciences). Ten thousand cells were analysed from each sample with FlowJo 10.9.0 software (BD Biosciences).

To monitor chromatin-bound MCM2 (Figs. 5 and 6; Appendix Fig. S6), cells were permeabilized with 0.2% Triton X-100 in PBS on ice for 2 min and washed with ice-cold PBS twice. The pre-extracted cells were fixed with 4% PFA for 20 min at 4 °C. The fixed cells were washed with BSA/PBS-T (1% BSA and 0.2% Tween 20 in PBS) and blocked with 5% normal goat serum in BSA/PBS-T for 30 min at RT. Subsequently, the cells were incubated with anti-MCM2 rabbit monoclonal antibody for 1.5 h at RT. After washing three times with BSA/PBS-T, the cells were incubated with anti-rabbit goat Alexa 647 antibody for 1 h at RT. Subsequently, the cells were washed three times with BSA/PBS-T and were incubated with 24 µg/mL of propidium iodide and 50 µg/mL of RNase A in 1% BSA/PBS for 30 min at RT. Fluorescent signals in the immuno-stained cells were detected by a BD Accuri C6 flow cytometer. We analysed more than 9000 cells (Figs. 5E, 6C, and 7A; Appendix Fig. S6E) or 3000 cells (Figs. 5F and 6D; Appendix Fig. S6F) for each sample with FlowJo 10.9.0 software. The background level of loaded MCM2 (black dots in Figs. 5E, 6C, and 7A; Appendix Fig. S6E) was determined by using a control sample without MCM2 staining. The gates for each cell cycle (G1, early S, late S, and G2/M) were manually defined.

### Immunoblotting

Cells were harvested by trypsin and washed with culture medium and PBS. Cells were lysed with RIPA buffer (25 mM Tris-HCl pH 7.5, 150 mM NaCl, 1% NP-40, 1% sodium deoxycholate, 0.1% SDS) containing complete protease inhibitor cocktail (Roche, #1187580001) for 30 min on ice. Subsequently, the tubes were centrifuged for 15 min at 4 °C and then the supernatant was mixed with the same amount of SDS-sample buffer (Cosmo Bio, #423420) before heating at 95 °C for 5 min. The denatured protein samples were separated on a 7.5% or 10% TGX Stain-Free gel (BioRad) and transferred to a nitrocellulose membrane (Cytiva, #10600003). The membrane was processed with 1% skim milk in TBS-T for 15 min and incubated with primary antibody at 4 °C overnight. After washing with TBS-T, the membrane was incubated with the corresponding secondary antibody for 1 h at RT. Proteins were detected by the ChemiDoc Touch MP imaging system (BioRad), and the detected signals were quantified using Image Lab software (ver. 6.0.1, BioRad).

### Antibodies

All antibodies used in this study are listed in Table EV3.

### Chromosome spreads

RAD21-mACl/SMC2-BTCh cells were seeded in a six-well plate. Two to three days later, cells were cultured in the presence and absence of 1 µM 5-Ph-IAA and/or 0.5 µM AGB1 for 2.5 h. Subsequently, KaryoMAX Colcemid Solution (Thermo Fisher Scientific, #15212012) was added to the medium at a final concentration of 0.1 µg/mL and then cells were incubated for an additional 0.5 h. Cells were harvested, washed with culture medium, and resuspended in 1 mL of pre-warmed 75 mM KCl before incubation at 37 °C for 15 min. Subsequently, 30 µL of MeOH/ acetic acid (3:1) was added to the tube, and incubated for 5 min at RT. After centrifugation, the supernatant was removed, and the cell pellet was resuspended with 500 µL of MeOH/AA and incubated for 5 min at RT. This incubation step was repeated with fresh MeOH/AA. After centrifugation, cells were resuspended in 150 µL MeOH/AA and incubated at –30 °C until observation. Five microlitres of the resuspended sample was dropped onto a tilted slide of grass and dried for up to 30 min at 60 °C. Dried nuclei were embedded with 5 µL of VECTASHIELD with DAPI (Vector Laboratories, #H-1200) in a coverslip.

### Microscopy

For immunofluorescence staining, cells were cultured on coverslips in the presence or absence of 1 µM 5-Ph-IAA and/or 0.5 µM AGB1 for the indicated time and fixed with 4% PFA for 20 min at RT. After washing with PBS-T, cells were permeabilized with 0.3% Triton X-100 in PBS for 5 min at RT. After washing, the cells were blocked with 5% BSA in PBS for 20 min at RT and incubated with an appropriate primary antibody for 1.5 h at RT. After washing, the cells were incubated with a secondary antibody and Hoechst 33342 for 1 h at RT. The coverslip with stained cells was mounted with 5 µL of ProLong Glass antifade mount (Thermo Fisher Scientific) on a slide glass. For detecting DNA replication by EdU incorporation, cells were cultured with 10 µM EdU for 30 min before fixation. Incorporated EdU was stained using the Click-iT EdU Imaging Kit (Thermo Fisher Scientific, #C10339) according to the manufacturer's instructions. The fluor-escent signals were captured with a Delta Vision Personal DV system (GE Healthcare) with an inverted microscope (IX71, Olympus) through a PlanApo 60×/1.42 oil immersion objective lens (Olympus). All pictures were deconvoluted. At least, 250 nuclei were processed for quantification of 53BP1 foci by using Volocity software (ver. 6.3.1, PerkinElmer). For observing mitotic spindles shown in Appendix Fig. S9B, images were taken by a FLUOVIEW FV3000 confocal microscope (Olympus) with a UPlanXApo 60x/1.42 oil immersion objective lens (Olympus).

### siRNA knockdown

We used dicer-substrate short interfering RNAs (DsiRNAs) (IDT). Two target sites each for ORC1 and CDC6 were used. Ten nM of DsiRNA was transfected using Lipofectamine RNAi MAX (Thermo Fisher Scientific, #13778075) following the manufac-turer's instruction.

### Quantitative RT-PCR

Total RNA samples were prepared by using RNAspin Mini (Cytiva, #25050071) following the manufacturer's instructions. Quantitative PCR were performed using One Step TB Green PrimeScript RT-PCR Kit II (Takara, RR086A) with Light Cycler (Roche). To detect mRNA, we designed the following primers: 5'-AACACAATG GACCTGCCAGAG-3', 5'-AGACAGTGCTGCTACCTTCCT-3' for ORC1 and 5'- TCCACTGGCGTCTTCACC-3', 5'-GGCAGAGAT-GATGACCCTTTT-3' for GAPDH. GAPDH was used as a control for calculating the ΔCt value.

## Data availability

This study includes no data deposited in external repositories.

The source data of this paper are collected in the following database record: biostudies:S-SCDT-10_1038-S44319-024-00224-4.

## Peer review information

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

## Acknowledgements

We thank all members of the Kanemaki laboratory for their discussion and support. We also thank Ms Karina Polkovnychenko, Dr Yasukazu Daigaku, and Dr Conner Craigon for the initial study as a NIG intern, for sharing the hTERT-RPE1 cell line expressing OsTIR1(F74G) and for providing a plasmid encoding BromoTag, respectively. We appreciate Dr Kazuhiro Maeshima and Ms Shiori Iida for the discussion on chromosome architecture. YH is a JSPS Research Fellow for Young Scientists (DC2), and MI is a MEXT scholarship fellow. This work was supported by JSPS KAKENHI (JP21H04719 and JP23H04925) and JST CREST (JPMJCR21E6) to MTK.

## Author contributions

**Yuki Hatoyama**: Conceptualization; Data curation; Formal analysis; Investigation; Methodology; Writing—original draft. **Moutushi Islam**: Conceptualization; Data curation; Formal analysis; Investigation; Methodology; Writing—original draft. **Adam G Bond**: Resources. **Ken-ichiro Hayashi**: Resources. **Alessio Ciulli**: Conceptualization; Resources; Writing—review and editing. **Masato T Kanemaki**: Conceptualization; Resources; Supervision; Funding acquisition; Investigation; Methodology; Writing—original draft; Project administration; Writing—review and editing.

Source data underlying figure panels in this paper may have individual authorship assigned. Where available, figure panel/source data authorship is listed in the following database record: biostudies:S-SCDT-10_1038-S44319-024-00224-4.

## Disclosure and competing interests statement

AC is a scientific founder, shareholder, and advisor of Amphista Therapeutics, a company that is developing targeted protein degradation therapeutic platforms. The Ciulli laboratory receives or has received sponsored research support from Almirall, Amgen, Amphista Therapeutics, Boehringer Ingelheim, Eisai, Merck KaaG, Nurix Therapeutics, Ono Pharmaceutical, and Tocris-Biotechne. The remaining authors declare no competing interests.

