## [Peer Review File · EMBO Reports]

Combination of AID2 and BromoTag expands the utility of degron-based protein knockdowns

Yuki Hatoyama, Moutushi Islam, Adam Bond, Ken-ichiro Hayashi, Alessio Ciulli, and Masato Kanemaki

Corresponding author(s): Masato Kanemaki (mkanemak@nig.ac.jp)

Review Timeline:

Submission Date:	6th Mar 24
Editorial Decision:	12th Apr 24
Revision Received:	25th Jun 24
Editorial Decision:	17th Jul 24
Revision Received:	23rd Jul 24
Accepted:	24th Jul 24

Editor: *Esther Schnapp*

Transaction Report:

Dear Prof. Kanemaki,

Thank you for the submission of your manuscript to EMBO reports. So far, I could only secure 2 referees for it and we have now received the enclosed reports from both of them. Given that they are in fair agreement, I am making a decision on your ms now based on the 2 reports we have and in the interest of time.

As you will see, both referees acknowledge that the findings are interesting. Both only have a few more suggestions for how the manuscript could be further improved and strengthened and I think all suggestions are good and should be addressed. Please let me know if you disagree or have any questions or comments. We can also discuss the exact revision requirements in a video chat, if you like.

I also paste below some comments from an advisor I contacted before sending your ms for peer-review. Please also consider this comment in your revised ms text.

I would thus like to invite you to revise your manuscript with the understanding that the referee concerns must be fully addressed and their suggestions taken on board. Please address all referee concerns in a complete point-by-point response. Acceptance of the manuscript will depend on a positive outcome of a second round of review. It is EMBO reports policy to allow a single round of major revision only and acceptance or rejection of the manuscript will therefore depend on the completeness of your responses included in the next, final version of the manuscript.

We realize that it is difficult to revise to a specific deadline. In the interest of protecting the conceptual advance provided by the work, we recommend a revision within 3 months (13th Jul 2024). Please discuss the revision progress ahead of this time with the editor if you require more time to complete the revisions.

- 1) A data availability section providing access to data deposited in public databases is missing. If you have not deposited any data, please add a sentence to the data availability section that explains that.
- 2) Your manuscript contains statistics and error bars based on $n=2$. Please use scatter blots in these cases. No statistics should be calculated if $n=2$.

3) We replaced Supplementary Information with Expanded View (EV) Figures and Tables that are collapsible/expandable online. A maximum of 5 EV Figures can be typeset. EV Figures should be cited as 'Figure EV1, Figure EV2' etc... in the text and their respective legends should be included in the main text after the legends of regular figures.

5) a complete author checklist, which you can download from our author guidelines <https://www.embopress.org/page/journal/14693178/authorguide>. Please insert information in the checklist that is also reflected in the manuscript. The completed author checklist will also be part of the RPF.

6) Please note that all corresponding authors are required to supply an ORCID ID for their name upon submission of a revised manuscript (<https://orcid.org/>). Please find instructions on how to link your ORCID ID to your account in our manuscript

tracking system in our Author guidelines

<<https://www.embopress.org/page/journal/14693178/authorguide#authorshippinguidelines>>

10) Regarding data quantification (see Figure Legends:

<https://www.embopress.org/page/journal/14693178/authorguide#figureformat>)

I look forward to seeing a revised form of your manuscript when it is ready.

Yours sincerely,

Referee #1:

In this manuscript, Hatoyama et al use 3 different degron tagging systems, either individually to degrade a reporter, or in combination to degrade proteins involved in higher order chromosome structure or DNA replication. They show that

- Individual degron tags sometimes fail to degrade proteins known to be essential to a sufficient degree to reveal the null phenotype
- Different degron tags, and different ligands for the same degron tag, can degrade a reporter protein with different efficiencies, and with profoundly different efficiencies of recovery
- Different degron tags can be used to exert orthogonal control over two distinct proteins
- Different degron tags fused to the same protein can act synergistically to induce more complete degradation, revealing null phenotypes
- Tandem or triple mAID tags greatly reduce the steady-state expression of fusion proteins in a Tir1-independent manner
- Cells simultaneously depleted of ORC1 and CDC6 enter mitosis without undergoing DNA replication.

The experiments are well performed and the conclusions, although in some cases not entirely novel or surprising, are justified. The authors should address the following comments before publication.

- 1) The slow recovery following dTAG- and BromoTAG-mediated depletion is a potentially significant finding of importance to understanding PROTAC pharmacology as well as informing researchers on which degron tag system to use. This section therefore deserves to be developed further. The authors should repeat the recovery experiment using the dTAG-13 ligand, for which efficient reversibility has been demonstrated by others (e.g. Nabet et al Nature Chem Bio 2018, Abuhashem et al Dev Cell 2022). Longer exposure times could be used to drive more complete dTAG-13-mediated degradation than shown in Figure 2.
- 2) The authors explanation for poor recovery is that the PROTAC ligands may remain associated with their substrate receptors to provide a washout-resistant ligand pool. If true, this pool might be reduced and reversibility might be more efficient if degradation is first achieved using lower ligand doses, over longer exposure times if necessary. This should be tested - can recovery be improved?
- 3) In Figure 6, it is stated that the double and triple mAID tag destabilises the target protein, but only steady-state expression data are presented. Can the authors show via proteasome inhibition or similar that the 2mAID and 3mAID-ORC1 fusions are less stable than the single tag?
- 4) Work demonstrating a very similar concept - combining AID with another degron tag in order to reveal null phenotypes via more complete degradation - was published some time ago by the Hochegger and Lindqvist groups (PMID: 32350921, PMID: 30008317, PMID: 34972955), including one of the same target proteins (CDC6) as the current study. Although one of these papers is cited (line 329), the authors should more explicitly acknowledge this prior work on double degrons.
- 5) Line 80 "Because IMiDs inevitably induce off-target proteolysis of CRBN neo-substrates". This statement may need to be updated in light of the recent Science paper from David Liu's group

Referee #2:

This is a timely and interesting study by Hatoyama et al. that covers two important aspects that are of interest to the wider cell biology community. Firstly the paper gives a detailed quantitative comparison of the most up-to-date induced deflation systems and sets a precedent for the use of double mAID-Bromo degron tags. Secondly, the study addresses an important question in the DNA replication field: Is Orc1 an essential replication licensing protein? Using the double degron method the authors show that only enhanced depletion of Orc1 causes a complete block in proliferation but even in these circumstances origins still manage to fire albeit in an erroneous manner.

Both points are important and I am confident that this MS will be well received by many researchers interested in the degron technology, as well as by the DNA replication field.

The study is technically sound and the experiments are mostly well performed and documented.

I only have a few minor comments and one suggestion for an additional experiment. Other than that I would support the publication of this study in EMBO reports

Minor comments:

1) The Lemmens et al. study that is cited in the discussion sets an actual precedent for this study both in terms of findings on Cdc6 requirements for DNA replication, uncoupling of S and M-phase and double degron tags (in this case Smash-AID) enhancing degradation. I think it would be fair to discuss the Lemmens paper in the introduction in the context of this study rather than bury this paper in a brief comment in the discussion.

2) A technical comment: FACS data and colony assays should be quantified from data from at least 3 repeats. For example, in Figure 5E, what is the proportion of S-phase cells? How reproducible is this across three biological repeats? Likewise, median points of three independent experiments should be shown in the MSM loading assays (E.g Figure 5F) and p-values should be calculated on the three median points rather than the large number of single-cell data that generate very small p-values for small differences in the data.

3) Suggested experiment: I'm still puzzled by the fact that even the strongest possible depletion of Orc1 does not abolish DNA replication. Are a few molecules of Orc1 enough to support licensing, or is there a true parallel mechanism involving Cdc6 and Orc1? This will be hard to determine, but the authors should try and further enhance Orc1 depletion by combining mRNA degradation using siRNA and depletion of the already synthesised protein with the double degron. This will reduce synthesis rates as well as enhance degradation and ultimately cause maximum depletion of Orc1. Will this be sufficient to cause a complete block in replication?

Advisor's comment:

The effect of co-depleting ORC1 and CDC6 is interesting to me as a member of the replication field, as the data are so clear, yet the interpretation is complicated by the fact that the phenotype comes from depleting two proteins, so one can't conclude whether ORC1 is absolutely essential for MCM loading (it almost certainly is) and the challenge is just to deplete it sufficiently well, or whether the key thing is to deplete both ORC1 and CDC6 (e.g. if other pathways for MCM loading were to exist, though until now no such pathway has been identified).

Our response to Reviewers and Advisor

(The comments are shown in blue, while our responses are shown in black.)

Response to Reviewer 1

We sincerely thank this reviewer for giving constructive comments and raising the issues to improve the manuscript.

1) The slow recovery following dTAG- and BromoTAG-mediated depletion is a potentially significant finding of importance to understanding PROTAC pharmacology as well as informing researchers on which degron tag system to use. This section therefore deserves to be developed further. The authors should repeat the recovery experiment using the dTAG-13 ligand, for which efficient reversibility has been demonstrated by others (e.g. Nabet et al Nature Chem Bio 2018, Abuhashem et al Dev Cell 2022). Longer exposure times could be used to drive more complete dTAG-13-mediated degradation than shown in Figure 2.

Following your comments, we tested whether more prolonged exposure to dTAG-13 would drive more depletion. This was indeed the case (Figure S2C). We also carried out the recovery experiment with 100 nM dTAG-13 after 12 h exposure and found that the recovery was not good (see Figure for Reviewer 1, panel A). We did not include this data because we tested recovery data with a lower concentration of 5-Ph-IAA, dTAGv-1 and AGB1, as you will see in our following response.

2) The authors explanation for poor recovery is that the PROTAC ligands may remain associated with their substrate receptors to provide a washout-resistant ligand pool. If true, this pool might be reduced and reversibility might be more efficient if degradation is first achieved using lower ligand doses, over longer exposure times if necessary. This should be tested - can recovery be improved?

We carried out experiments and added new data (Figure S3B). In these experiments, we treated the reporter cells with the DC50 concentration of each ligand identified in Figure 2B. After 12 h treatment, the GFP expression level became approximately 30%, and we started the recovery from this time point. As you predicted, the recovery became better than shown in Figure 2D. However, the overall trend was the same: The recovery with AID2 was better than the other two.

3) In Figure 6, it is stated that the double and triple mAID tag destabilises the target protein, but only steady-state expression data are presented. Can the authors show via proteasome inhibition or similar that the 2mAID and 3mAID-ORC1 fusions are less stable than the single tag?

Thanks for pointing it out. We also wanted to understand the reason why 2mAID and 3mAID lowered the expression level of ORC1. So, we investigated the half-life of the fusion protein by adding cycloheximide and the mRNA level by qPCR (Figure S7C, D). We found that 3mAID-ORC1 has a shorter half-life compared to mAID- and 2mAID-ORC1 (Figure S7C). We are surprised to see that the mRNA level of 2mAID-ORC1 was significantly reduced (Figure S7D), suggesting the insertion of 2mAID to the N-terminal coding region of the *ORC1* gene affected transcription or mRNA

stability. Both 2mAID and 3mAID reduced the expression of ORC1, but the mechanism might differ. We wrote these findings in the main text (lines 379-383 on page 16).

4) Work demonstrating a very similar concept - combining AID with another degron tag in order to reveal null phenotypes via more complete degradation - was published some time ago by the Hochegger and Lindqvist groups (PMID: 32350921, PMID: 30008317, PMID: 34972955), including one of the same target proteins (CDC6) as the current study. Although one of these papers is cited (line 329), the authors should more explicitly acknowledge this prior work on double degrons.

We agree that we should have acknowledged their prior work on AID-SMASH. So, we changed Introduction accordingly and added citations (lines 104-106 on page 5).

One thing that we noted was that the AID-SMASH system did not work for CDC6 in HCT116 cells, even though we used the same tagging construct that came from Dr. Lindqvist (Figure for Reviewer 1, panels B and C). It seemed that a cleaved CDC6 proteins without the mAID-SMASH tag was produced (shown by red arrow, panel B) in the cells, and this protein was not degraded and supported cell growth (panel C). We wanted to compare our AID2-BromoTag with AID-SMASH, but we could not do it.

5) Line 80 "Because IMiDs inevitably induce off-target proteolysis of CRBN neo-substrates". This statement may need to be updated in light of the recent Science paper from David Liu's group

Thanks for letting us know. We changed the text accordingly in Introduction and cited the paper (lines 81-83 on page 4).

Response to Reviewer 2

We are delighted to read your comments stating, "I am confident that this MS will be well received by many researchers interested in the degron technology, as well as by the DNA replication field". We appreciate you taking the time to evaluate our manuscript and provide us with constructive comments and questions.

1) The Lemmens et al. study that is cited in the discussion sets an actual precedent for this study both in terms of findings on Cdc6 requirements for DNA replication, uncoupling of S and M-phase and double degron tags (in this case Smash-AID) enhancing degradation. I think it would be fair to discuss the Lemmens paper in the introduction in the context of this study rather than bury this paper in a brief comment in the discussion.

Thanks for pointing it out. Please read our response to point 4 from Reviewer 1.

2) A technical comment: FACS data and colony assays should be quantified from data from at least 3 repeats. For example, in Figure 5E, what is the proportion of S-phase cells? How reproducible is this across three biological repeats? Likewise, median points of three independent experiments should be shown in the MSM loading assays (E.g Figure 5F) and p-

values should be calculated on the three median points rather than the large number of single-cell data that generate very small p-values for small differences in the data.

We changed the figures showing the loading level of MCM and added statistical information. Please look at Figures 5F, 6D, S6F and S7E.

3) Suggested experiment: I'm still puzzled by the fact that even the strongest possible depletion of Orc1 does not abolish DNA replication. Are a few molecules of Orc1 enough to support licensing, or is there a true parallel mechanism involving Cdc6 and Orc1? This will be hard to determine, but the authors should try and further enhance Orc1 depletion by combining mRNA degradation using siRNA and depletion of the already synthesised protein with the double degron. This will reduce synthesis rates as well as enhance degradation and ultimately cause maximum depletion of Orc1. Will this be sufficient to cause a complete block in replication?

We thought that the raised issue was important. So, we decided to conduct the suggested experiments (Figure S8A). By combining AID2-BromoTag with siRNA, we depleted ORC1 and CDC6 to near the detection limit by WB (Figure S8B). In both cases, the combination of siRNA and AID2-BromoTag achieved more pronounced defects in S phase progression (Figure S8C, D), showing that MCM loading still occurred when ORC1 or CDC6 was depleted only by AID2-BromoTag (Figures 5E and S6E). Intriguingly, the combinational depletion of CDC6 induced almost complete suppression of DNA replication (Figure S8D), while that of ORC1 still allowed some DNA replication (Figure S8C). These data suggest two possibilities, which are hard to clarify: there was still ORC1 existing even after the combinational depletion, or MCM-loading occurred to some extent in the absence of ORC1. However, please note that ORC1 is pivotal to achieving MCM loading enough to drive cell proliferation. In this regard, ORC1 is essential for cell viability. In contrast, CDC6 is likely absolutely essential for MCM-loading. We added texts in Results and Discussion accordingly (lines 412-426 on page 17 and 537-540 on page 22).

Response to Advisor's comments

The effect of co-depleting ORC1 and CDC6 is interesting to me as a member of the replication field, as the data are so clear, yet the interpretation is complicated by the fact that the phenotype comes from depleting two proteins, so one can't conclude whether ORC1 is absolutely essential for MCM loading (it almost certainly is) and the challenge is just to deplete it sufficiently well, or whether the key thing is to deplete both ORC1 and CDC6 (e.g. if other pathways for MCM loading were to exist, though until now no such pathway has been identified).

Thanks for pointing out this important issue. Please read our comments to point 3 from Reviewer 2 and take a look at Figure S8.

A**B****C****Figure for Reviewer 1**

(A) The HCT116 reporter cells were treated with 100 nM dTAG-13 for 12 h. Subsequently, the cells were washed and the recovery of GFP was monitored up to 24 h. (B) Comparison of CDC6-mAID-SMASH with the other double-degrons. We generated HCT116 CDC6 degron cell lines by fusing mAID-BromoTag (mAB), mAID-SMASH (mAS), 2mAID or 3mAID to the C-terminus of CDC6. CDC6 levels in cells treated with the indicated ligands. In the CDC6-mAID-SMASH cells, a smaller CDC6 protein shown by red arrow was not degraded. Because we confirmed that the both CDC6 alleles are tagged with mAID-SMASH, it is likely that the mAID-SMASH tag was cleaved after translation. (C) The CDC6-mAID-SMASH cells did not show growth defects in when treated with the inducing ligands.

Dear Prof. Kanemaki

Thank you for the submission of your revised manuscript. We have now received the enclosed reports from the referees and I am happy to say that both support its publication now.

Only a few more minor editorial requests will need to be addressed before we can proceed with the official acceptance of your manuscript:

- Please upload the final ms file without any figures.
- Please correct the conflict of interest subheading to "Disclosure and Competing Interests Statement"
- We received an automatic note for Adam Bond about no longer being at ICR. Please send us his current email with the final ms file.
- We need from you a completed author checklist, which you can download from our author guidelines <<https://www.embopress.org/page/journal/14693178/authorguide>>. The completed author checklist will also be part of your transparent peer-review process file.
- The funding info on Young Scientists (DC2) and MEXT scholarship are missing in our online ms submission system, please add it with the final submission.
- Please upload the 3 tables as EV tables (Table EV1, etc) and correct their names in the table files and in the manuscript callouts.
- The APPENDIX FILE needs a table of content with page numbers; and the nomenclature (and the ms callouts) needs to be corrected to Appendix Figure S1, etc.
- The source data need to be uploaded as one (zipped) folder per figure and can have single files per figure panel.
- Materials & Methods should be called just Methods.
- During our figure check of accepted ms, we found a possible reuse within Figure 5B / Tubulin (left side): 5-Ph-IAA & AGB1. The blots are very similar in the figure. Blots are different in the source data than in the figure. Can you please check your figure/source data and explain what happened and redraw the figure if necessary.
- Please add the exact p values in the legends of figures 5f; 6d.
- Although 'n' is provided, please describe the nature of entity for 'n' in the legends of figures 2b; 3d; 4a-b.

I made a few minor changes to the abstract that needs to be written in present tense. Please let me know whether you agree with the following:

Acute protein knockdown is a powerful approach to dissecting protein function in dynamic cellular processes. We previously reported an improved auxin-inducible degron system, AID2, but recently noted that its ability to induce degradation of some essential replication factors, such as ORC1 and CDC6, was not enough to induce lethality. Here, we present combinational degron technologies to control two proteins and enhance target depletion. For this purpose, we initially compare PROTAC-based degrons, dTAG and BromoTag, with AID2 to reveal their key features and then demonstrate control of cohesin and condensin with AID2 and BromoTag, respectively. We develop a double-degron system with AID2 and BromoTag to enhance target depletion and accelerate depletion kinetics and demonstrate that both ORC1 and CDC6 are pivotal for MCM loading. Finally, we show that co-depletion of ORC1 and CDC6 by the double-degron system completely suppresses DNA replication, and the cells enter mitosis with single-chromatid chromosomes, indicating that DNA replication is uncoupled from cell cycle control. Our combinational degron technologies will expand the application scope for functional analyses.

EMBO press papers are accompanied online by A) a short (1-2 sentences) summary of the findings and their significance, B) 2-3 bullet points highlighting key results and C) a synopsis image that is exactly 550 pixels wide and 200-600 pixels high (the height is variable). The synopsis image should provide a sketch of the major findings, like a graphical abstract. Please note that text needs to be readable at the final size. Please send us this information along with the final manuscript.

Kind regards,
Esther

Referee #1:

The authors have added data and text to their revised manuscript which improves the quality, and addresses all of my concerns. Congratulations on a nice piece of work.

Referee #2:

The author has addressed my concerns and produced interesting new results. The paper should now be published and will be an important contribution to cell biology.

The authors have addressed all minor editorial requests.

Prof. Masato Kanemaki
National Institute of Genetics
Department of Chromosome Science
Yata 1111
Mishima 411-8540
Japan

Dear Prof. Kanemaki,

I am very pleased to accept your manuscript for publication in the next available issue of EMBO reports. Thank you for your contribution to our journal.

Yours sincerely,
